# EMERGENCE OF COLLECTIVE POLICIES INSIDE SIMULATIONS WITH BIASED REPRESENTATIONS

## ABSTRACT

We consider a setting where biases are involved when agents internalise an environment. Agents have different biases, all of which resulting in imperfect evidence collected for taking optimal actions. Throughout the interactions, each agent asynchronously internalises their own predictive model of the environment and forms a virtual simulation within which the agent plays trials of the episodes in entirety. In this research, we focus on developing a collective policy trained solely inside agents' simulations, which can then be transferred to the real-world environment. The key idea is to let agents *imagine together*; make them take turns to host virtual episodes within which all agents participate and interact with their own biased representations. Since agents' biases vary, the collective policy developed while sequentially visiting the internal simulations complement one another's shortcomings. In our experiment, the collective policies consistently achieve significantly higher returns than the best individually trained policies.

## 1 INTRODUCTION

We look at the world through our own lenses. Two people with the same exact observations create different internalizations of the world through their own mental models, collect different amounts of evidence, take different actions, and end up with different outcomes. For example, students attending a lecture look at different parts of the presentation slides, and doctors looking at the clinical log of a patient focus on different data and statistics and make different diagnoses.

In model-based reinforcement learning (RL) (Kaelbling et al., 1996; Sutton & Barto, 1998), agents interact with an environment through models that are analogous to the human mental models (Hamrick, 2019). However, just as our decisions governed by our mental models are often far from perfect, the agents' policies that are designed upon the models are suboptimal (Box, 1976; 1979), as the inductive biases used to learn such models are not guaranteed to be correct (Griffiths et al., 2010).

In this research, we propose a learning algorithm of a *collective policy* that leverages multiple agents' biased representations encoded from their internal models. Like humans who often compensate for their cognitive biases through collaboration (Woolley et al., 2010; Muchnik et al., 2013), the goal of the collective policy is to make use of multiple internal models so that the biased representation formulated by each internal model complements one another to achieve higher returns than the best-performing individual policy. The collective policy in this research has the following characteristics:

— *Asynchronous shaping of agents' internal models:* The collective policy operates on agents' preestablished internal models. Prior to learning the collective policy, agents construct their internal models over the course of their own interactions with the environment, which can be done asynchronously and independent of other agents. Note that this learning process of the collective policy does not fall into the conventional multi-agent RL settings, where agents are concurrently situated in the same environment (Littman, 1994; Panait & Luke, 2005; Bu et al., 2008; Buşoniu et al., 2010).

— *Collective policy learned solely inside agents' internal simulations:* In model-free and model-based RL, policy learning typically requires a large amount of interaction with the environment, which hampers feasibility, time-efficiency (Guez et al., 2018), and safety (Garcıa & Fernández, 2015). This is particularly prohibitive in our setting, as during the collective policy learning, we need to force the agents to go through the same rollouts together, coordinate their actions, and interact with the environment in algorithmically intended ways (see e.g., Yahya et al. (2017); Rubenstein et al. (2014);

Nagpal (2016)). To overcome this, we propose a learning algorithm of the collective policy that involves zero interaction with the real environment; instead, the collective policy is trained solely inside agents' simulations, analogous to human's mental simulation or imagination (Hamrick, 2019). In our learning algorithm, agents take turns to be the host that generates a virtual rollout, within which all agents participate in and learn the collective policy together.

— *Minimal number of parameters to learn the collective policy:* In the proposed algorithm, the number of parameters and the training iterations of the collective policy function is same as those of the individual policy functions. This not only allows the efficient training, but also assures that the performance edge of the collective policy over the individual policies is not attributable to the model complexity but to the complementing evidences provided by the multiple internal models. Moreover, learning of the collective policy can precede learning of the individual policies.

We make the following contributions in this paper:

   I. We present a novel problem setting in model-based RL that resembles the human cognitive bias of imperfect information. Note that our setting is clearly different from the partial observation settings in RL (Astrom, 1965; Kaelbling et al., 1998); in our setting, two agents at the same exact time and location in the environment encode different representations from the same observation.

  II. We compare the real environment and the internal simulation generated by agent models for training a policy. With appropriate early stopping epoch and $\tau$, we can achieve much better scores at a fraction of the training time.

 III. We present a learning algorithm of the collective policy that operates on the simulations generated from internal models of agents and complements their biased representations, thereby achieving higher scores than that of the best-performing individual policy.

 IV. We experiment with VIZDOOM, a different environment in which we observe an unexpected behavior of agents becoming delusional, i.e., having the false belief about the reward system, preventing us from learning the collective policy in simulation.

  V. We release our source code, visual demonstrations, additional results, and in-depth discussions in a separate link [1].

## 2 COLLECTIVE POLICY TRAINING ALGORITHM

Here, we describe the process of training a collective policy by leveraging multiple agents' internal models that create their own biased representations given the same observation as inputs. We start by describing the components of agents' internal models and explain how each agent creates its own biased representation of an observation inside the model. Then, we introduce additional components required for training the collective policy inside the agents' simulations.

### 2.1 INTERNAL MODELS OF AGENTS

We use world models (Ha & Schmidhuber, 2018a;b; Schmidhuber, 2015) as agents' internal models. World model abstracts spatio-temporal features of the environment in a compact representation that is used for policy learning after the model training. The left panel of Figure 1 illustrates the two components of the world model along with the environment and the controller C that learns the policy after the world model is trained. First, at each time step $t$, the visual sensory component V abstracts a high dimensional input observation from the environment into a compressed representation. Variational autoencoder (VAE) (Kingma & Welling, 2014; Rezende et al., 2014) is used to encode a 2D image input frame into a latent vector $z_t$. Then, the latent vector $z_t$ as well as the following action $a_t$ are used as inputs for the memory component M, which outputs a prediction of the next encoding $z_{t+1}$. Mixture-density recurrent neural network (MD-RNN) (Bishop, 1995; 1994; Graves, 2013; Ha & Eck, 2018; Graves, 2015) is used for this component for the stochastic prediction of the future, i.e., $P(z_{t+1}|z_t, h_t, a_t)$, where $h_t$ is the hidden state variable at time $t$. V and M use sequential image frame data from multiple rollouts from a random policy and apply variational inference and backpropagation algorithm for inference and learning.

After the two components of the world model are trained, the parameters of C are optimised online during the following rollouts using covariance matrix adaptation evolution strategy (CMA-ES) (Hansen

---

[1] http://s000.tinyupload.com/?file_id=54935721167326296555

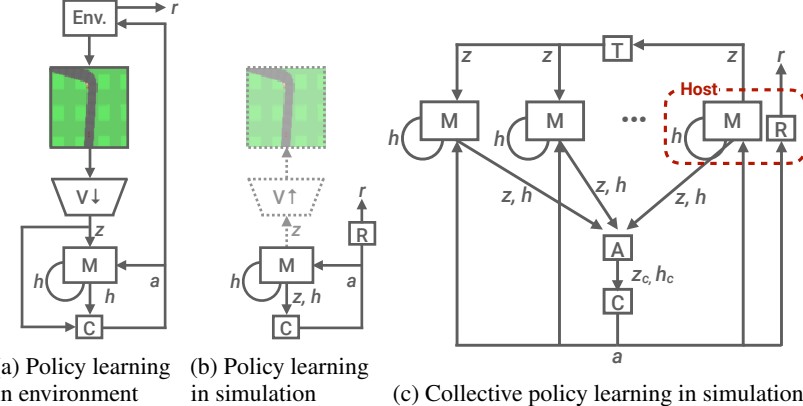

(a) Policy learning    (b) Policy learning
in environment         in simulation              (c) Collective policy learning in simulation

Figure 1: Flow diagrams of policy learning of (a) an individual policy while interacting with an environment, (b) individual policy learning inside internal simulation, (c) collective policy learning involving multiple agents' internal models. Note that for (b) and (c), there is no interaction with the environment. For collective world model, we omit the decoding part of V for clear presentation.

& Ostermeier, 2001; Hansen, 2006; 2016) which is shown to be the method-of-choice for small solution spaces up to few thousand parameters (Salimans et al., 2017; Loshchilov & Hutter, 2016).

**Training policy inside an agent's simulation**    The memory component M predicts the latent representation of the next frame and allows an agent to take an action based on the agent's prediction of the near future as well as the current situation. Further, one can generate multiple future frames, even the whole episode from start to finish (Figure 1b). Then, C is trained inside this virtually generated environment (Ha & Schmidhuber, 2018a;b). MD-RNN, which is the realisation of M, can endow a desirable level of uncertainty in the simulation by adjusting the temperature parameter $\tau$.

**Biased representations**    We hypothesize that when an agent's model internalizes the observation through V, the agent from collecting the full evidence needed to learn the optimal policy because of bias. The bias, which persists throughout the experiment, can be that certain parts of the frame are left out (occlusion), or only a certain part of the frame is accessible (magnification). [2]

**Asynchronous shaping of agents' world models**    Prior to our merging of agents' simulations, we assume that agents preestablish their internal models (M and V) asynchronously, based on their imperfect evidence accumulated. However, it is necessary to ensure there are environment states that all agents share, e.g., few frames that all agents co-observe, to train the translator component. This will be further explained in the following subsection. Unlike the typical distributed ML systems (Chen et al., 2012; Deisenroth & Ng, 2015; Low et al., 2015), each internal model can be trained independently without broadcasting or establishing the communication protocol among agents, i.e., each agent's encoded representations from V can be done using independent VAEs, each with different initialisations. This is because the collective controller and all individual controllers are trained independently of each other Cs, as well as the remaining components V and M.

### 2.2    Components for learning the collective policy

We gather up agents' biased representations created inside the individual world models and train a collective policy in the internal simulations generated from the agent models so that each model's biased representation complements others' shortfalls and the virtual return per episode is maximised in a collective manner. Here, the key idea is to make agents be in the same internal simulation at the same time, where each agent's model takes a turn to be the *host* of the virtual rollout using her respective memory component M (Figure 1c). Throughout this section, we suppose that there are a total of $N$ internal models in a system, and the hosting agent per virtual rollout is denoted by $H$. Added components for the collective policy learning are explained as follows:

---

[2]By design, our algorithm can operate on different kinds of biases, e.g., temporal distortions, although we did not empirically test the performance under such conditions.

**Rule component (R)**   When learning a policy in the environment, we automatically retrieve reward signals from the environment. However, when the (collective) policy is trained in simulation, we need to learn how the reward is given using previous and current state representations. Within each agent's model, we only preserve the memory component M and introduce an additional component R that understands the rule of the reward system. For every virtual rollout we have a hosting agent $H$ whose rule component $R_H$ returns a virtual score which is instantly transferred to the controller component C so that C can assess the quality of the current policy solely inside simulation and without the need to interact with the actual environment. For this component, we use RNN that takes previous and current biased representations as inputs and returns the estimated reward for each time step.

**Translator component (T)**   In our setting, agents' visual sensory components create different representations from the same observation. While we train the collective policy inside agents' simulations, the translator component T converts biased representations between two agents. For every virtual rollout generated by the hosting agent $H$, we need $N - 1$ translators $T_{H,a}$ for $a \in \{1, 2, \ldots, N\} \backslash \{H\}$, which are the functions $T_{H,a}(z_H) = z_a$. In our setting, the internal representations are much smaller in dimensionality than the original observations, and representations are highly intercorrelated among different internal models. Therefore, we apply meta-learning (Finn et al., 2017; 2018; Kim et al., 2018; Nichol et al., 2018) to speed-up the training of T and reduce the training data.

**Aggregator component (A)**   To train a collective policy, there is an aggregator component A which is a permutation-invariant function that takes representations from agents' internal models as inputs and returns their summarisations of the same shape. Here, we take the summation function as an aggregator which simple but effective (Ali et al., 2018; Garnelo et al., 2018a;b; Soelch et al., 2019).

---

**Algorithm 1:** Collective policy learning algorithm

---

**Input:** $N$ trained memory components components M

1 $\quad\quad\quad$ $N$ trained rule components R

2 $\quad\quad\quad$ $N^2$ trained translator components T

3 Initialize parameters for the collective policy function;

4 **while** *not converged* **do**

5 $\quad$ Set $RewardTotal \leftarrow 0$;

6 $\quad$ **for** *hosting agent $a = 1, \ldots, N$* **do**

7 $\quad\quad$ Set the hosting agent $H \leftarrow a$;

8 $\quad\quad$ Retrieve an initial encoding for the first time step $z_{H,0}$;

9 $\quad\quad$ **for** *time $t$ until episode terminates* **do**

10 $\quad\quad\quad$ Generate encodings of other agents $z_{a,t}, \forall a \in \{1, 2, \ldots, N\} \backslash \{H\}$ using $z_{H,t}$ from T;

11 $\quad\quad\quad$ Generate $h_{a,t}$ for all agents using their respective M and aggregate $N$ concatenated vectors $[z_{a,t}; h_{a,t}]$ using A to return $[z_{c,t}; h_{c,t}]$;

12 $\quad\quad\quad$ Insert the aggregated vector$[z_{c,t}; h_{c,t}]$ in the controller C to get an action at time $t$;

13 $\quad\quad\quad$ Execute the action inside the hosting agent's simulation M to infer the next encoding $z_{H,t+1}$ and to get a reward $r$ from R;

14 $\quad\quad$ **end**

15 $\quad\quad$ Set $RewardTotal \leftarrow RewardTotal + r$;

16 $\quad$ **end**

17 $\quad$ Update the parameters for the collective policy using CMA-ES and $RewardTotal$;

18 **end**

---

**Training scheme for C**   We use CMA-ES that operates on a state space to optimise the parameters of C. To use the simulations of all hosting agents, for each training iteration, we define a single episode by concatenating multiple virtual rollouts and use the sum of the cumulative rewards calculated from the hosting agent's rule component $R_H$. Aside from making use of all agents' simulations, combining multiple rollouts as a single episode is shown to stabilise policy learning throughout the training (Ha, 2017). We highlight that the number of parameters for C as well as the number of training iterations are identical to those of the individual controllers.

For each iteration, the controller component C operates on exactly the same input space as the individual model as we take the summation function and $|[z_c; h_c]| = |[z_a; h_a]|, \forall a \in \{1, \ldots, N\}$.

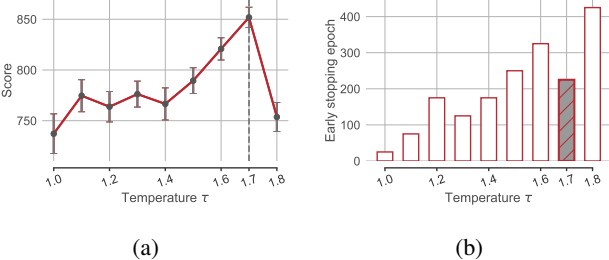

(a)          (b)

Figure 2: (a): Relationship between the temperature parameter $\tau$ of MD-RNN in internal simulation and the score received from the real environment. $\tau$ governs the stochasticitiy of the virtual environment generation, and there is an optimal value, 1.7 that gives the best score. (b): As $\tau$ increases, the simulated environment becomes more unpredictable and we accordingly need more training epochs.

Table 1: Comparison between training the policy inside real environment and inside an agent's internal simulation. $\pm$ is followed by one standard error.

| | # Env. interaction | | # Epoch | Time / epoch (min.) | Pop. size | Mean | Median |
|---|---|---|---|---|---|---|---|
| | Model tr. | Policy lr. | | | | | |
| Env. | 10 K | 1.84 M | 1,800 | 42 | 64 | $900.0 \pm 5.2$ | 911.9 |
| Sim. | 10 K | **0** | 225 | **0.8** | 32 | $851.9 \pm 9.8$ | 875.1 |

Therefore, the only added terms in the time complexity of collective world model come from T and A, where for T, we need $N - 1$ forward passes of RNNs and A that aggregates $N$ input vectors.

In algorithm 1, we summarise how the learning of the collective policy works inside agents' simulations using trained components M, R, and T along with the aggregator A.

## 3 EXPERIMENT

In this section, we answer the following questions: First, how does training a policy in internal simulation compare to training a policy in the real environment? What specifications do we need for comparable performance in the simulation? Second, how much do we gain from the collective policy trained within the simulations of agents' internal models whose biased representations complement one another? What causes the collective policy to perform better than the individual policies? In addition, we introduce another environment where when bias is involved during the internalization, agents in the environment become *delusional*, not being able to understand the rule of the environment.

### 3.1 EXPERIMENTAL SETUP

Agents learn to drive in OpenAI gym CARRACING environment (Brockman et al., 2016; Klimov, 2016). Each trial consists of 1,000 image frames, and while agents are driving a car, the score is calculated as the total number of tiles, i.e., the gray area in Figure 3a, the car visited. Thus, agents need to keep moving forward without derailing from the track until the trial ends in order to maximise the return. For each time step, agents can take three continuous actions: steer left/right and accelerate. Throughout the experiment, we report an average score over 100 trials with one standard error.

Following the previous setting by Ha and Schmidhuber (Ha & Schmidhuber, 2018a;b), we train each internal model's components using 10,000 rollouts with random policy. As displayed in Figure 3a, we make each agent's observations imperfect by 1) occluding a certain part of the frame or 2) only showing a magnified portion of the frame. Here, we conserve the car image (in red) for all agents' observations, as this image portion is critical for the rule component to calculate the virtual score. Also, we highlight that this data-gathering phase is the only time interacting with the real environment throughout the experiment, except for the testing trials. Finally, we keep the same number of parameters and maximum training iterations for both individual and collective policies.

### 3.2 EXPERIMENTAL RESULTS

**Policy learning in simulation**  Figure 2 describes how the temperature parameter $\tau$ of MD-RNN (M component) affect the policy learning inside simulation. As $\tau$ increases, the virtual environment

Table 2: Results of the collective policy learning. When multiple agents' internal models provide complementary evidences from their biased representations, the collective policy achieves higher real-environment scores than that of the best performing individual internal model.

| Bias type | # M | Overlap (%) | Cov. (%) | Indiv. policy | | | Collective Policy |
|---|---|---|---|---|---|---|---|
| | | | | Worst | Mean | Best | |
| Occ. | 2 | 0 | 100 | $615.2 \pm 22.7$ | 618.4 | $621.6 \pm 7.9$ | $\mathbf{750.8 \pm 12.3}$ |
| Mag. | 2 | 12.5 | 37.5 | $494.3 \pm 16.5$ | 558.7 | $623.0 \pm 22.9$ | $\mathbf{733.8 \pm 19.9}$ |
| Mag. | 3 | 18.8 | 37.5 | $494.3 \pm 16.5$ | 602.1 | $688.7 \pm 17.4$ | $\mathbf{744.6 \pm 10.2}$ |
| O.,M. | 4 | 100 | 100 | $494.3 \pm 16.5$ | 581.0 | $623.0 \pm 22.9$ | $\mathbf{742.1 \pm 08.8}$ |
| Occ. | 2 | 100 | 62.5 | $468.4 \pm 21.9$ | 541.8 | $615.2 \pm 22.7$ | $629.1 \pm 15.5$ |
| Mag. | 2 | 100 | 39.1 | $571.9 \pm 21.4$ | 678.3 | $784.7 \pm 9.3$ | $769.8 \pm 16.3$ |

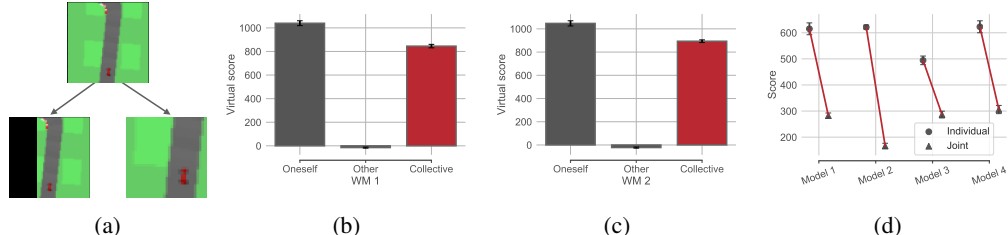

|  (a)  |  (b)  |  (c)  |  (d)  |

Figure 3: (a): Original observation (top) with decodings of the biased representations of different internal models (occlusion and magnification). (b) – (c): Virtual scores when the policy is trained inside its own internal simulation (left bars), when using the policy that is trained in other's internal simulation (center bars), and when the collective policy is used (right bars). If other's policy is used in the virtual environment generated by oneself, the virtual score decreases catastrophically, whereas the collective policy gives reasonable scores inside all simulations. (d): "Joint" is the collective policy without T and A components; when the collective policy is trained without co-observation, it performs worse than individual policies.

generated by the internal model becomes more unpredictable. In Figure 2a, we observe that there is an optimal temperature parameter, which is 1.7 in CARRACING environment, that gives the score when the policy learned in simulation is transferred to the real environment. This result is in accordance with the result reported by Ha & Schmidhuber (2018a), where they experimented in the VIZDOOM environment. In Figure 2b, we observe that there is an additional trend that incorporates $\tau$; With all $\tau$, we observe that when we train the policy in simulation for too long, the policy overfits the virtual environment so that the virtual score monotonically increases as the training epoch increases but performs worse when the policy is transferred to the real environment. Thus, the early stopping is required when training the policy in simulation, and the early stopping epoch generally increases as we increase $\tau$, meaning that we need more training iterations until we transfer the policy to the real environment when the virtual environment is more unpredictable.

From Table 1, We report that with an appropriate early stopping epoch and $\tau$, we can achieve a score of $851 \pm 9.8$ in CARRACING environment with the policy that is trained only in simulation and then transfered to the real environment, compared to the score of $900.0 \pm 5.2$ when trained in the real environment. When the policy is trained in simulation, the training time per epoch is approximately 53 times faster, total train time changes from 52 days to 3 hours, and the number of rollouts generated in the environment is reduced from 1.84 million to zero during the policy training. [3]

**Collective policy** Table 2 shows the results of the collective policy compared with the individual policies whose internal models that are integrated. Here, #M is the number of internal models used for training, Overlap is the maximum percentage of frame overlaps between agents' biased representations in terms of pixel count, and Cov. is the percentage of total coverage when all biased representations are superimposed in terms of pixel count. From the first four rows in Table 2, we

---

[3] To our knowledge, there is no previous research that trained policies for a single task in both real and virtual game environments; in the original world models paper, the CARRACING experiment is done in the actual environment, and the VIZDOOM experiment is done inside simulation, but not the other way around.

Figure 4: (a): Biased representation (occlusion) in VIZDOOM environment. (b) – (e): 4 different agents' internal models with the bias setting given in (a). Although trained under exactly the same setting, internal models (b) and (d) learns reasonable policies inside simulation, while (c) and (d) became delusional. For all four internal models, their virtual scores are similar (blue bar), but the actual score significantly varies between normal and delusional agents. Also, once an agent's model is trained, delusional agents do not become normal during the policy training phase and vice versa.

confirm that when we train the collective policy that integrates multiple internal models whose biased representations complement one another, we can achieve higher scores than that achieved by the best-performing individual policy in the real environment. In the last two rows, we do not observe any significant increase in the score when the collective policy is compared with the individual policies, because one internal model's biased representation is a proper superset of that of the other's.

Figures 3b – 3c show the virtual scores of individual and collective policies. When an individual policy is trained inside the simulation generated by its own model, we achieve the best score. However, when the trained policy is executed in the simulation generated by another model, the score decreases catastrophically. In comparison, the collective policy returns reasonable scores in all simulations, although the score is less then the one achieved by the individual policy in its own simulation.

Figure 3d highlights the necessity of the model structures proposed in this research for learning the collective policy. Here, "Joint" corresponds to real-environment score achieved by the collective policy jointly trained by sequentially visiting multiple internal simulations, but while doing so, the agents other then the host does not participate in the simulation. In other words, its the collective policy without the T and A components. From this result, we confirm that the main performance edge of the collective policy comes from the "co-observation" during the policy learning.

**Delusional agents** We report that in the VIZDOOM environment (Paquette, 2016; Kempka et al., 2016), an environment where the rules of the game and the other aspects of the environment are tightly coupled, agents randomly become *delusional* when the bias is involved in their internal models (4). Here, the rule that governs the reward system of the game is highly intertwined with physics of the system (movement and perspective of the fireball, explosion of the fireball when the ball contacts the agent) and is potentially affected by every pixel of the frame, and therefore the rule component R inevitably has to be incorporated within the memory component M (this phenomenon is extensively studied by Ellefsen et al. (2019)). In this case, some agents' internal model will falsely believe that they will be rewarded not by dodging the fireball but by going inside the occluded region and stay there until the trial ends. To our knowledge, this type of delusion has previously not been observed; in the world models paper, agents will sometimes *cheat* inside simulation by creating an easy virtual environment when the temperature parameter is set to a low value but clearly it is a different type of behaviour. This abnormal behaviour is remarkable in two ways. First, an agent falling into a delusion happens completely at random at the training phase of the memory component. Thus, for the same exact training data, methods, and iterations, some agent will behave normally (treating the distortion as part of the environment) while some will be delusional. Second, once the memory component is trained and the agent shows delusional behaviour, it can never get out of this however many times we train the policy with different initialisations. On the other hand, in the car racing task, this issue can be resolved because the reward system is only governed by a regional area surrounding the car; therefore, if we do not distort this part and train R and M separately, we can prevent agents from falling into this type of delusion. In our collective setting, this issue is particularly problematic because, for delusional agents, the degree of agents' bias do not directly transfer to their performances.

## 4 RELATED WORK

Machine learning (ML) and RL researchers propose internal models of the agents that is analogous to humans' mental models (see the work of Hamrick (2019) for the review), and the concept of learning a policy solely inside an agent's simulation using recurrent structures was proposed by Schmidhuber

in 1990s (Schmidhuber, 1970; 1991; 1990) and recently been reviewed in the name of *learning to think* (Schmidhuber, 2015; Lake et al., 2017). Also, Racanière et al. (2017) develop a neural architecture that leverage internalisations of the world on top of the real-environment interactions, and Ha & Schmidhuber (2018a;b) propose world models that enable full operationalisation inside simulation. Additionally, the idea of internalising the world is adopted for meta-control and to optimise and computation based on task difficulty and compexity (Hamrick et al., 2017; Pascanu et al., 2017). Finally, there is a recent approach in model-based RL that further develops and applies the idea of simulation or imagination to achieve better results. Nair et al. (2018) and Hafner et al. (2019) make use of agents' internal models to pre-determine the long-term outcomes (e.g., imagined outcomes) and develops a policy based on such outcomes.

In this research, the environments that we experiment on has state spaces that are accommodable for the internal models to capture with random policies. Orthogonal to our research, Hafner et al. (2019) advances the idea of training a policy inside a simulation by proposing an end-to-end framework that incorporates simulations in environments that cannot be fully explored and described by random policies, and by doing so, balances out the optimizations of the exploration policies and the model learning while minimizing the number of interactions with the environment.

Close to our research in terms of conceptual proximity and motivation is a study by Bard et al. (2019), In their work, authors focus on a board game called Hanabi in which agents operate in a cooperative fashion to solve a task. Similar to our work, agents in the Hanabi game only has access to imperfect information, and their information can be complemented only by interacting with other agents. Here, the most critical diverging point is that the Hanabi challenge seeks an optimal policy for each individual, which ideally mimics human's sophisticated behaviours such as the theory of mind (Premack & Woodruff, 1978), and the authors do so by interacting with the actual environment. On the other hand, we focus on designing a specific model structure that can be solely operationalised in multiple agents' simulations and deriving a collective policy that gives better performance than individual ones. On a related note, Battaglia et al. (2018) proposes a general framework that discovers structural, relational, and compositional representations of an arbitrary system that builds upon graph networks and complements each entity's inductive biases, which can be applied to our setting with more entities and with relational information.

## 5 DISCUSSIONS

We postulate a scenario in which each agent's internalisation of the world is biased, resulting in imperfect evidence collected at each timestep. Our goal is to simultaneously superimpose agents' biased representations portrayed by their internal models to learn a collective policy so that agents' defective simulations can be complemented by others. In our experiment, we used the CARRACING platform with a maximum of four agents and showed that the collective policy with integrated simulations give better performance for the car driving task. However, a few outstanding questions remain that need further exploration to better comprehend the overall process of learning a collective policy inside simulation and make it more effective and scalable.

The first direction is to find an optimal group composition that maximises the score of the collective world model when the number of agents is fixed. Coordinating an ideal group is a non-trivial task because the collective performance depends not only on each individual's ability but also on the degree of each agent's feature complementing one another (Kim et al., 2016) and the needs for optimal coordination among agents will become more salient in massive settings (Suarez et al., 2019). In our task, we tried to investigate the relationship between the number of total pixels covered by the group of agents and the performance but were unsuccessful at finding any significant correlations. The second approach is improving the model architecture to overcome the performance degrade, especially in the aggregator component. We conjecture that our current setting of naively superimposing internal representations might potentially "dilute" useful bits of information as the number of agents grows. Recently, simple aggregator architectures, such as summation or mean functions have been implemented in previous research (Ali et al., 2018; Garnelo et al., 2018a;b) and extended by adopting e.g., attention mechanisms (Bahdanau et al., 2015; Xu et al., 2015; Wang et al., 2018; Vaswani et al., 2017) as by Kim et al. (Kim et al., 2019) or tower representation architectures by Eslami et al. (Ali et al., 2018). Finally, we tried finding the complete observation "frame-wise" (rather than directly finding the optimal collective policy), with agents' scores and their encodings at every time step and using Bayesian optimisation (Mockus, 2012; Shahriari et al., 2015; Močkus, 1975) with neural processes (Garnelo et al., 2018a;b; Kim et al., 2019) as the surrogate function, but was unsuccessful because of time complexity.

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

## A  LINKS FOR VIDEO TRIALS AND SIMULATIONS

In supplementary material, we attach videos for actual trials and simulations.

### A.1  CARRACING

**CarRacing collective trial**  $carracing\_collective\_1.gif$ and $carracing\_collective\_2.gif$ files are the real-environment trial using collective policy with 2 agents' internal models. Since we cannot visualise the decoding of the superimposed biased representations, we present the viewpoint of each agent. The performance of this collective policy is reflected in the main draft.

**CarRacing individual trial**  $carracing\_individual\_1.gif$ file is the real environment trials using individual internal model.

### A.2  VIZDOOM

We demonstrate the *delusion* of agents in simulation under biased representations, which is reported and discussed in section 3.2.

**Normal agent in simulation**  $vizdoom\_normal\_agent.mp4$ file is simulation of normal agent with occlusion.

**Delusional agent in simulation**  $vizdoom\_delusional\_agent.mp4$ file is simulation of delusional agent with occlusion. These two agents are trained with exactly the same specification. We tried the same specification for not just these two agents, but with 6 more, confirmed that this delusion occurs randomly.

## B  ADDITIONAL RELATED WORK

**Nascent development of simulation in ML**  In reinforcement learning (RL) (Kaelbling et al., 1996; Sutton & Barto, 1998), the concept of simulation develops from prediction. Model-based RL builds a predictive model that abstracts key features of observations and leverage the state space to predict future observations for planning. For example, there is a line of work that uses such predictive powers of the world for short-step lookaheads using FNNs (Guzdial et al., 2017) and RNNs (Sutton, 1990; Werbos, 1987; Dosovitskiy & Koltun, 2017), while another branch focuses on generating longer-range predictions conditioned on time step-wise actions using raw pixel image frames as inputs, but without policy learning (Chiappa et al., 2017; Oh et al., 2015; Leibfried et al., 2017; Denton & Birodkar, 2017; Watters et al., 2017). Model-free RL also benefits from exploiting the predictive power by directly predicting future rewards or values using deep neural nets (Silver et al., 2017; Oh et al., 2017; Tamar et al., 2016).

**Humans' efficacy of imagination**  For humans, the benefit of imagination, in the names of mental practice, mental training and cognitive rehearsal, is empirically demonstrated in varying fields of learning e.g., musical instruments (Meister et al., 2004), physical actions (Premack & Woodruff, 1978), and social interactions (Decety & Grèzes, 2006; Lateef, 2010) and validated with neurological (Cisek & Kalaska, 2004) and cognitive (Driskell et al., 1994) evidence.

**Distributed ML**  Finally, in terms of model architecture, our research falls into the branch of distributed ML, and specifically, into recently proposed model fusion (Hoang et al., 2019a) and collective machine learning (Hoang et al., 2019b). Similar to ours, Hoang et al. focuses on a setting where there are multiple independent experts, and the objective is to uncover the communication protocol among them in a distributed manner to reach the consensus prediction. However, the target domains of this line of research and other recent advances in distributed machine learning (Feraud et al., 2019; Qu et al., 2019) mainly lie in privacy and security (e.g., a setting where each expert treats confidential information) or reducing the overload of the central server. On the other hand, in our research, the communication protocol issue among agents' internal models is structurally negated since although we train each agents' V component (VAE) independently, there's a collective

controller C that's also trained independently to all the individual V components and also to the individual C components.

## C    TRAINING SPECIFICATIONS

We used the same setting as in the original world models paper for training the individual components V and M.

### C.1    CONTROLLER COMPONENT C

For C, we used CMA-ES for training. We concatenated 16 rollouts to define one episode, and each virtual rollout is hosted by an agent in the collective system. We made sure that the number of rollouts in the episode can be exactly divided by the total number of agents in collective world model with remainder of 0, so that all agents can be hosts the same number of times. Also, with this setting, we manage to keep the training iteration of collective policy exactly the same with the individual policies. For example, if there are 8 agents in total, each can host simulations 2 times in a single episode; if there are 4 agents in total, each can host simulations 4 times in a single episode, and so forth.

We set the population size to 32, and each population is assigned with a single CPU. We set the initial sigma value of CMA to 0.1, and set all solution values to 0 for initialisation. Finally, for every training, we used random seed numbers.

### C.2    RULE COMPONENT R

For the rule component R, we used RNN with 2 hidden layers, each layer with the same size of 256. We used Adam as optimizer, with initial learning rate of 0.001, and used mean squared error for the loss function. We also used dropout for all hidden layers with the value 0.1, gradient clipping value of 0.25, batch size of 100, and training epochs of 500. For inputs, for every time step $t$, we used generated encodings of the current as well as the 10 previously generated encodings, so in total of 352-dimensional input space was used, and scalar value of the instant reward was returned.

Here, we note that in our setting, the model complexity of R is generously set. Since training of this component was not the bottleneck of our computation, we did not try to shrink the model architecture, as only focused on the model returning low training/testing errors. Therefore, the current model size might be overly large, and we believe that this size can be drastically reduced without significant loss in accuracy.

### C.3    TRANSLATOR COMPONENT T

For the trainslator component T, we used FNN with 4 hidden layers, each layer with sizes 1024, 512, 256, and 128. We used Adam as optimizer, with initial learning rate of 0.001, and used mean squared error for the loss function. We also used dropout for all hidden layers with the value 0.1, batch size of 10,000, and training epochs of 250. Inputs and outputs are both 32-dimensions, which corresponds to the size of the encoding $z$.

As in the rule component R, we believe that model size of T can also be drastically reduced.

### C.4    COMPUTING INFRASTRUCTURE

Our model is programmed using python Tensorflow.

Training C is done in multi-CPU environment, as CMA-ES can be effectively implemented in parallel computations. We used 32 CPU cores for training both collective and individual C, where each core is assigned with a single population in CMA-ES.

For training V (VAE), M (MD-RNN), T(FNN), R(RNN), we used a single GPU GTX-1080 Ti for training each component.

