# OpenReview forum: "Emergence of Collective Policies Inside Simulations with Biased Representations"
_ICLR.cc/2020/Conference — Reject_

### Official Review · AnonReviewer3 · 2019-10-10
**Official Blind Review #3**

**Rating:** 3

**Review:**

The paper studies the settings where multiple agents each act independently in different copies of an environment, without interacting with each other. Each agent uses model-based learning, learning a representation of the world, then learning a controller against the learned world model (learned with an evolutionary strategy of CMA-ES). Different agents will naturally learn world models with different biases - this paper proposes a method that learns a collective policy from the biased models of the different agents, by letting each agent observe imagined rollouts from other agents' world models, training on that data. They find this improves performance over learning within each agent's individual model.

From a style perspective, I found this paper hard to read. Many terms are immediately abbreviated into non-conventional single letter abbreviations (C for controller, T for Translator). This extends to the results table - it's quite difficult to understand the results in Table 1 and Table 2 without a careful reading to find what what the abbreviations mean, and I still don't understand what the Overlap (%) and Cov. (%) columns mean in Table 2, despite reading the paragraph explaining it several times. The analysis section also makes a distinction between delusional agents and cheating agents, which I also didn't understand. The GIFs provided demonstrate that the model moves towards the occluded region, and this is described as a novel delusion that hasn't been observed before, distinct from the cheating behavior from Ha and Schmidhuber's world models paper. I don't understand why these are different - in both cases, the agent travels to a region of space where no fireballs appear, and since reward is defined as not getting hit by a fireball, this is optimal behavior for the imperfect world model.

To created biased representations, the authors apply cutout or zoom in on specific parts of the state. I found this surprising - shouldn't different models naturally lead to biased representation by themselves? The VAEs used should naturally be inaccurate enough to lead to different imperfections.

There are very few details for the translator T, that translates latent z_i from world model M_i into the correspond latent x_j for world model M_j.  This seems like a very important detail of the method, I could see different translators T leading to very different results, and would have like to see ablations for how different Ts affected the training procedure.

Overall, the question I found myself asking the entire paper was, "what does this add on top of existing work on training against an ensemble of world models?" This question has been studied in previous work (a quick search turned up https://arxiv.org/abs/1809.05214 and https://arxiv.org/abs/1802.10592). In these prior works, instead of learning a translator T between representations, we decode an imagined rollout from the VAE for each world model, in the original high-dimensional input state (images, in the case of this paper), and simply train on that data instead. This does not require learning O(N^2) translator models translating z_i to z_j for each pair of world models. I would have liked to see a comparison to this approach, because right now I don't see why you would want to learn translators T in the first place. Simply aggregating rollouts across models would still lead to individually biased world models that can be aggregated to learn a single controller that performs better than any single model.

Edit: have read author reply, no changes to review.

**Experience Assessment:**

I have read many papers in this area.

**Review Assessment: Checking Correctness Of Derivations And Theory:**

I assessed the sensibility of the derivations and theory.

**Review Assessment: Checking Correctness Of Experiments:**

I assessed the sensibility of the experiments.

**Review Assessment: Thoroughness In Paper Reading:**

I read the paper at least twice and used my best judgement in assessing the paper.

---

### Official Review · AnonReviewer1 · 2019-10-22
**Official Blind Review #1**

**Rating:** 1

**Review:**

This paper presents a model-based RL architecture based on World Models which additionally makes use of an ensemble of models. Specifically, each model in the ensemble is independently trained models and has different amounts and types of occlusions (i.e. part of the image cropped out or a small piece of the image is magnified). The ensemble of models are used to train what is referred to as a “collective” policy in the CarRacing and VizDoom environments. The collective policy achieves better performance than policies trained using the individual models and achieves higher data efficiency than model-free RL.

While I think the paper tackles an interesting question---how to leverage multiple models of the world that may be trained on different data and with different capabilities---I unfortunately do not feel that this paper is ready for publication at ICLR, and thus recommend rejection. My justifications are that (1) the paper suffers from clarity issues and several cases makes claims or statements which are incorrect, (2) the proposed evaluation is somewhat contrived and the method does not seem to generalize well to other domains, and (3) there is not much engagement with the rest of the literature on model-based RL. I think that addressing #1 and #3 alone would substantially improve the paper. An additional change that would make the paper much more compelling would be to train the models on more ecologically valid observations, such as from multiple modalities or different camera angles (see below for discussion).


My first concern with the paper has to do with clarity. Overall, I found the paper quite difficult to read as it uses a lot of unusual terminology to explain what it does. For example, the paper uses “rule component” to refer to what would normally be a “reward function”. Similarly, although this is an ensemble method, the term “ensemble” is never used even once. The word “bias” is also used in a way which is different from how it is normally used---here it seems to refer to the fact that the agents see different things, whereas in ML bias typically has a precise meaning in terms of the bias-variance tradeoff. I also noticed several places throughout the paper where citations or related work seem to be misunderstood:

- 1st page, 2nd paragraph: this is not what Griffiths (2010) is about, and is not really an appropriate use of the term “inductive bias”
- 2nd page, contribution 1: the paper claims that it is not dealing with a partially observable setting, but this is exactly what the paper is doing. Specifically, each model in the ensemble receives different partially observed observations from the environment. Although in traditional RL partially observed usually means “one agent, one observation of a state”, there is nothing limiting the general formulation of partial observability to cover the case of “multiple agents, multiple observations of the same state”.
- 8th page, line 2: Lake et al (2017) is not really about “learning to think”.
- Section 4, last paragraph: this seems to be not very relevant to this paper, as the present paper is not about a multiagent system or about graph-based representations.

There are some additional issues with clarity of the model explanation; for example, it’s not clear how the translator component is trained (the paper states that it is trained with meta-learning, but this is vague).

My second concern is that the approach is somewhat contrived and does not seem to generalize well to other domains. In particular, under what circumstances would you want to train multiple models that receive inputs that are missing different pieces? I can see the appeal of learning different models in some cases (e.g. different camera angles, or different sensory modalities) but these have not been evaluated here which makes the whole thing seem a bit arbitrary. Moreover, while it seems to work ok in the CarRacing domain (though the ensemble-trained policy does not perform as well as a non-ensemble agent which is trained with full observations) it does not seem to work well in the VizDoom environment. On the basis of these two issues it is not at all clear to me how well the proposed method will work in more ecologically valid settings requiring multiple models.

Finally, my third concern is that the engagement with the rest of the model-based RL literature is lacking. As mentioned above, this method is an ensemble method but there is no discussion of other model-based RL ensemble methods (e.g. [1] and [2]). Similarly, the paper dismisses other work on partial observations but this work is quite relevant to the present paper which does indeed include partial observations (see above). While there is a lot of text spent discussing other work such as literature from cognitive science, this is not really that relevant to the present paper and that space would be better spent discussing related work from model-based RL.

Some additional minor comments:

It would be interesting to see a comparison between the ensemble policy trained with models that always receive the same (full) observations versus the present approach (where they receive partial observations). This might still provide some benefit as the models will still result in slightly different latent representations due to different initialization and training data, so it would be interesting to see whether this helps as well even if the observations are the same.

What does “model tr” and “policy lr” mean in Table 1?

In the various tables and figures, how are the error bars computed? Are they over episodes, or agent seeds? In general, how many seeds were used to perform evaluations? (Ideally it would be at least 3).

Page 2, section 2.1: “V and M use sequential image frame data from multiple rollouts from a random policy and apply variational inference and backpropagation algorithm for inference and learning.” → this is imprecise; they do not apply both variational inference and backpropagation; backpropagation is used to implement variational inference.

I found the results of Figure 3d quite interesting, though am wondering if you controlled for the number of evaluations of each model? E.g. if you only train the policy on one model at a time versus training them on all models at a time, do you train for longer in the first case so that the policy ultimately receives the same amount of information from all the models?

[1] Kurutach, T., Clavera, I., Duan, Y., Tamar, A., & Abbeel, P. (2018). Model-ensemble trust-region policy optimization. arXiv preprint arXiv:1802.10592.
[2] Chua, K., Calandra, R., McAllister, R., & Levine, S. (2018). Deep reinforcement learning in a handful of trials using probabilistic dynamics models. In Advances in Neural Information Processing Systems (pp. 4754-4765).

**Experience Assessment:**

I have published in this field for several years.

**Review Assessment: Checking Correctness Of Derivations And Theory:**

N/A

**Review Assessment: Checking Correctness Of Experiments:**

I assessed the sensibility of the experiments.

**Review Assessment: Thoroughness In Paper Reading:**

I read the paper at least twice and used my best judgement in assessing the paper.

---

### Official Review · AnonReviewer2 · 2019-10-24
**Official Blind Review #2**

**Rating:** 3

**Review:**

This paper proposes a new approach to conducting RL: it proposes to first train a collection of individual agents independently, and then exploit them to learn one collective policy. Each agent has an observation space that is differently impaired, such as by blocking out some specific set of pixels in its image observations. Each such agent learns its own dynamics and reward models based on its own observations, and goes on to train a policy based on simulated rollouts using these learned models. In the second phase, a collective policy is trained as follows: every training episode happens inside a different "host" agent's simulation, and the collective policy is thus effectively trained on this mixture of variously inaccurate simulations. Since different individuals have different observations, the collective policy receives an input observation that is the aggregate of all agents' individual observations. Throughout, this paper builds on top of the "world models" approach proposed in Ha & Schmidhuber 2018.

Most sections of this paper are well-written and the high-level ideas are novel and interesting. My main issues with the paper are listed below, in rough order of priority:
(1) I find the experimental evaluations short of convincing.
  * Baselines: The proposed collective policy is evaluated against the individual policies of agents in its population, which is a very weak baseline since the individual agents are artificially impaired. It is also evaluated against an upper bound: a  policy trained directly in the real world, and shown to be only slightly weaker in performance. I would suggest the following additional baseline:
     - individual agents without observation impairments, trained in simulation (essentially the world model paper this approach builds on). ["world models"]
     - the average of the individual policies ["policy ensemble''].
     - a policy trained with a model that is a composite average of the individual world models ["model ensemble"]. This is the in the spirit of prior work on model ensembling, such as Chua et al 2018, " ... handful of trials."
  * Environments: The results presented so far indicate the proposed approach works on one environment (CarRacing) and does not on another (VizDoom). I would like to see experiments on more environments to figure out whether CarRacing is the exception or the rule.
  * choice of observations for individual agents: in a general setting, how are these to be engineered?

(2) It does not sufficiently acknowledge or clarify connections to other work in the field, such as on model bootstrapping in model ensembles (e.g. Chua et al 2018, "... handful of trials.") or evaluate against them. In particular, I think the proposed approach is a novel way to achieve model bootstrapping, plus some additional tweaks: rather than train different models on different subsets of training experience, it trains different models on different feature views of the same training episodes.

(3) It does not sufficiently motivate its setting: when is it true in realistic settings that it would make sense for different agents in an environment to be artificially impaired in different ways? Experiments are only conducted in contrived settings where portions of the environment are deliberately removed from the observation for different agents. I also have a somewhat related suggestion: perhaps future versions of the paper might focus on using different modalities (such as RGB images and depth) for the different agents.

(4) From my understanding, there seems to be a tradeoff between how much overlap there exists between different agents' observations and how translatable different agents' views are to each other. In other words, the method requires the translator T to take one view of an agent and transform it into (a feature representation of) the view of another agent. This seems ill-defined in general settings.

(5) Related to the above, how is the aggregated vector [z_{ct}, h_{ct}] generated for the collective policy when it is eventually evaluated in the real world? Are the features computed separately by each agent before aggregation, or is one agent selected and the translator T used again? If the former, then does this induce a domain shift between training time when the policy is trained on predicted features from the translator, and testing time when the policy sees true features corresponding to different agents?

**Experience Assessment:**

I have read many papers in this area.

**Review Assessment: Checking Correctness Of Derivations And Theory:**

N/A

**Review Assessment: Checking Correctness Of Experiments:**

I assessed the sensibility of the experiments.

**Review Assessment: Thoroughness In Paper Reading:**

I read the paper at least twice and used my best judgement in assessing the paper.

---

### Author Response · Authors · 2019-11-15
**Thank you all the reviewers for your comments.**

We appreciate your time and effort to review our paper. In particular, thank you for the comments about the contrived setting, missing crucial citations, and the translator component. Those comments will be very helpful when we improve our research in the future.

---

### Decision · Program_Chairs · 2019-12-19

**Decision:**

Reject

**Comment:**

This paper presents an ensemble method for reinforcement learning.  The method trains an ensemble of transition and reward models.  Each element of this ensemble has a different view of the data (for example, ablated observation pixels) and a different latent space for its models.  A single (collective) policy is then trained, by learning from trajectories generated from each of the models in the ensemble.  The collective policy makes direct use of the latent spaces and models in the ensemble by means of a translator that maps one latent space into all the other latent spaces, and an aggregator that combines all the model outputs.  The method is evaluated on the CarRacing and VizDoom environments.

The reviewers raised several concerns about the paper. The evaluations were not convincing with artificially weak baselines and only worked well in one of the two tested environments (reviewer 2). The paper does not adequately connect to related work on model-based RL (reviewer 1 and 2). The paper does not motivate its artificial setting (reviewer 2 and 1).  The paper's presentation lacks clarity from using non-standard terminology and notation without adequate explanation (reviewer 1 and 3).  Technical aspects of the translator component were also unclear to multiple reviewers (reviewers 1, 2 and 3).  The authors found the review comments to be helpful for future work, but provided no additional clarifications.

The paper is not ready for publication.